# Prognostic prediction models for clinical outcomes in patients diagnosed with visceral leishmaniasis: protocol for a systematic review

James Wilson  ,[1,2] Forhad Chowdhury,[1,2] Shermarke Hassan,[1,2] Elinor K Harriss,[3] Fabiana Alves,[4] Prabin Dahal,[1,2] Kasia Stepniewska,[1,2] Philippe J Guérin[1,2]

[1]Infectious Diseases Data Observatory (IDDO), University of Oxford, Oxford, UK
[2]Centre for Tropical Medicine and Global Health, University of Oxford, Oxford, UK
[3]Bodleian Health Care Libraries, University of Oxford, Oxford, UK
[4]Drugs for Neglected Disease Initiative, Geneva, Switzerland

**Correspondence to**
Dr James Wilson;
james.wilson@iddo.org

## ABSTRACT

**Introduction** Visceral leishmaniasis (VL) is a neglected tropical disease responsible for many thousands of preventable deaths each year. Symptomatic patients often struggle to access effective treatment, without which death is the norm. Risk prediction tools support clinical teams and policymakers in identifying high-risk patients who could benefit from more intensive management pathways. Investigators interested in using their clinical data for prognostic research should first identify currently available models that are candidates for validation and possible updating. Addressing these needs, we aim to identify, summarise and appraise the available models predicting clinical outcomes in VL patients.

**Methods and analysis** We will include studies that have developed, validated or updated prognostic models predicting future clinical outcomes in patients diagnosed with VL. Systematic reviews and meta-analyses that include eligible studies are also considered for review. Conference abstracts and educational theses are excluded. Data extraction, appraisal and reporting will follow current methodological guidelines. Ovid Embase; Ovid MEDLINE; the Web of Science Core Collection, SciELO and LILACS are searched from database inception to 1 March 2023 using terms developed for the identification of prediction models, and with no language restriction. Screening, data extraction and risk of bias assessment will be performed in duplicate with discordance resolved by a third independent reviewer. Risk of bias will be assessed using the Prediction model Risk Of Bias ASsessment Tool (PROBAST). Tables and figures will compare and contrast key model information, including source data, participants, model development and performance measures, and risk of bias. We will consider the strengths, limitations and clinical applicability of the identified models.

**Ethics and dissemination** Ethics approval is not required for this review. The systematic review and all accompanying data will be submitted to an open-access journal. Findings will also be disseminated through the research group's website (www.iddo.org/research-themes/visceral-leishmaniasis) and social media channels.

**PROSPERO registration number** CRD42023417226.

## STRENGTHS AND LIMITATIONS OF THIS STUDY

⇒ We present a protocol for a robust and comprehensive systematic review of visceral leishmaniasis (VL) prognostic models, using current best-practice guidelines on data extraction, risk of bias assessment and reporting.

⇒ Inclusion criteria are designed to identify a broad range of VL prognostic model studies, including all patients with a VL diagnosis, and with no exclusions based on treatment setting, type of clinical outcome or prediction horizon.

⇒ We describe a comprehensive and evidence-based search strategy to identify a broad range of prognostic model studies across five large bibliographic databases, with no limitations on language or initial publication date.

⇒ Unpublished, non-peer-reviewed studies, such as conference abstracts and educational theses, are not included in the eligibility criteria.

⇒ A systematic assessment of the current use and impact of VL prognostic models is considered outside the scope of the planned review.

## INTRODUCTION

Visceral leishmaniasis (VL), a parasitic infection transmitted between mammalian hosts via the bite of an infected sandfly, is a disease mostly prevalent in tropical regions that disproportionately touches vulnerable people affected by forced migration, malnutrition and poverty.[1] The disease often presents insidiously with fever, splenomegaly and weight-loss, and is almost always fatal without effective treatment.[2] The WHO estimates an incidence of 50000–90000 cases per year,[3] resulting in approximately 400000 disability-adjusted life years lost and over 5000 deaths.[4] However, accurate estimates of disease burden are obfuscated by limited country-level reporting, evolving dense foci of infection in remote areas, and a paucity of active surveillance.[1] Despite progress made

over the last 20 years, successful treatment remains challenged by high drug costs, prolonged treatment courses requiring hospitalisation and frequent drug side effects.[5] Patients with previous treatment failure or immunosuppressive comorbidities such as advanced HIV suffer from particularly high relapse and mortality rates.[1 6]

To optimise individual patient care and effectively balance the distribution of constrained resources, identification of patients at high risk of treatment failure and subsequent death is crucial. Risk stratification is also important on a population level; elimination programmes in endemic areas can use risk prediction tools to strategically target patients prone to treatment failure and hence reduce the infectious reservoir driving onward transmission. Prognostic prediction models (referred henceforth as prognostic models) play a central role in VL risk stratification; informing healthcare providers, policymakers and patients on the treatment setting, treatment regimen and intensity of follow-up.[7–9]

Systematic reviews of prognostic models have been published across a range of infectious diseases,[10–12] serving not only to inform healthcare providers on available risk stratification tools, but also as a research tool to identify candidate models for external validation or updating (recalibration) with data from new settings. Indeed, the lack of external validation studies is considered the greatest barrier to the broader acceptance of prognostic models as a reliable and acceptable clinical tool.[13–15]

## Aim

We will perform a systematic literature review to identify, summarise and appraise prognostic models in patients diagnosed with VL. Specifically, we focus on models that predict clinical outcomes such as treatment failure (initial failure or relapse) and death, developed subsequent to VL diagnosis.

The review will serve two principal purposes:

▶ Inform stakeholders, such as policymakers and healthcare workers directly involved in the treatment of VL patients, on the available prognostic models and their setting-specific clinical utility, strengths and limitations.
▶ Inform researchers interested in using their own data for the development, validation or updating of VL prognostic models.

## METHODS AND ANALYSIS

This systematic review will adhere to Preferred Reporting Items for Systematic Reviews and Meta-Analyses (PRISMA); a reporting guideline for systematic reviews,[16] and Transparent Reporting of a multivariable prediction model for Individual Prognosis or Diagnosis: checklist for Systematic Reviews and Meta-Analysis (TRIPOD-SRMA); a reporting guideline for systematic reviews of prediction models.[17]

Important protocol amendments will be documented on PROSPERO. We use the PRISMA Protocols

**Table 1** PICOTS approach to frame the research question

| | |
|---|---|
| Population | All patients with a confirmed or suspected diagnosis of visceral leishmaniasis as per study authors |
| Index model | All published prognostic models that develop, validate and/or update (recalibrate/extend) a risk model |
| Comparator model | Not applicable |
| Outcomes | Any clinical outcome that occurs following diagnosis |
| Timing | All prognostic models developed at the time of, or following diagnosis. No restriction on the prediction horizon |
| Setting | No restriction |

(PRISMA-P) checklist to guide reporting of the protocol (online supplemental material 1).[18]

## Study eligibility

We follow a PICOTS (Population, Index model, Comparator model, Outcomes, Timing, Setting) approach to frame our review question and inclusion criteria (table 1).[19 20]

Our population of interest includes all patients with a confirmed or suspected diagnosis of VL as reported by the study authors. We include all published, peer-reviewed studies that develop, externally validate, update, or any combination thereof, a prognostic model with the intention of predicting individual clinical outcomes following VL diagnosis.

In accordance with expert guidance on the methodology of prediction model research,[14 21 22] we define a prognostic model as a multivariable model (including two or more predictors) where the intention is to predict outcomes at the individual patient level. Prognostic model studies are distinguished from predictor finding or prognostic factor studies, where the intent is to investigate the effect of a single or group of factors on an outcome of interest.[23] We therefore exclude all studies that present models where the aim is not to predict risk at the individual patient level. We also exclude unpublished and non-peer-reviewed studies, including conference abstracts and educational theses, studies that only report diagnostic prediction models, and animal studies.

To complement the systematic review, using the same search strategy we will identify (1) systematic reviews and meta-analyses of prognostic models, and (2) any impact studies that investigate the clinical outcomes of using vs not using an eligible prognostic model. These studies will not be subject to formal data extraction, but will be summarised in a narrative review.

## Search strategy

An information specialist (EKH) created the search strategies to retrieve relevant records from the following databases: Ovid Embase; Ovid MEDLINE; the Web of Science Core Collection, SciELO and LILACS. The databases

**Table 2** Search strategy initially developed in Ovid Embase and searched on 1 March 2023 (search queries were subsequently adapted to other bibliographic databases)

| Query No. | Query terms |
|---|---|
| 1 | visceral leishmaniasis/ |
| 2 | ((Leishmaniasis and Visceral) or (Leishmania and infantum) or (Leishmania and donovani) or (Kala and azar)).ti,ab,kw. |
| 3 | 1 or 2 |
| 4 | Validat$.ti,ab. or Predict$.ti. or Rule$.ti,ab. or (Predict$ adj2 (Outcome$ or Risk$ or Model$)).ti,ab. or ((History or Variable$ or Criteria or Scor$ or Characteristic$ or Finding$ or Factor$) adj2 (Predict$ or Model$ or Decision$ or Identif$ or Prognos$)).ti,ab. or (Decision$ adj2 (Model$ or Clinical$)).ti,ab. or (Prognostic adj2 (History or Variable$ or Criteria or Scor$ or Characteristic$ or Finding$ or Factor$ or Model$)).ti,ab. |
| 5 | statistical model/ |
| 6 | decision*.ti,ab. |
| 7 | 5 and 6 |
| 8 | 4 or 7 |
| 9 | (Stratification or ROC Curve or Discrimination or Discriminate or c-statistic or c statistic or Area under the curve or AUC or Calibration or Indices or Algorithm or Multivariable).ti,ab. |
| 10 | receiver operating characteristic/ |
| 11 | 8 or 9 or 10 |
| 12 | 3 and 11 |

were searched from inception to 1 March 2023 with no language restriction. Where necessary, records published in languages not spoken by the authors will be translated using the Google Translate service (http://translate.google.com), or otherwise using a professional translation service. The search strategy used text words and relevant indexing terms to retrieve studies describing eligible prognostic models. The Ingui search filter was combined with an additional search string developed by Geersing *et al*,[24 25] and adapted for Ovid Embase (table 2) and remaining bibliographic databases (online supplemental material 2).

### Selection process and data extraction
All references were exported to Covidence for deduplication and screening (Covidence systematic review software, Veritas Health Innovation, Melbourne, Australia. Available at www.covidence.org).[26] Google Scholar will be used as a grey literature source once screening is complete to identify additional relevant studies. Citation searching of all included studies will be performed to identify further studies for screening.

Studies identified from the search strategy are being independently screened by two reviewers (JW, FC). Preliminary screening is at title and abstract level followed by full-text screening. A third reviewer (PD), an experienced statistician, will make the final judgement on study inclusion if discordance remains following discussion between the two screening reviewers. A flow diagram will be presented as per the PRISMA 2020 checklist.[16]

Study information will be extracted, collected and managed using the REDCap electronic data capture tools hosted at the University of Oxford.[27] A data extraction form will be created based on CHecklist for critical Appraisal and data extraction for systematic Reviews of prediction Modelling Studies (CHARMS) for data extraction and Prediction model Risk Of Bias ASsessment Tool (PROBAST; https://www.probast.org/) for assessing risk of bias (table 3).[20 22] A pilot form will be trialled prior to data extraction. Two independent reviewers (JW, SH), both with expertise in prediction modelling, will independently extract the study information. Where discordance remains after discussion, a final decision will be made by a third expert reviewer (PD). Study authors will not be contacted to obtain information not reported in the study.

### Risk of bias
Risk of bias will be summarised and reported separately for each model development and external validation using PROBAST.[22] Two reviewers (JW, SH), will answer 20 signalling questions across four domains (participants, predictors, outcome and analysis), which will be used to judge the overall risk of bias as either 'low', 'high' or 'unclear'. Discordance between the two reviewers will be resolved by a third reviewer (PD).

If impact studies are identified, their risk of bias will be assessed using the Cochrane Risk of Bias tool for randomised comparative designs or the Risk Of Bias In Non-randomised Studies of Interventions tool for studies using a non-randomised comparative design.[28 29]

### Data synthesis
We will present a narrative synthesis of our review findings. Key extracted information will be summarised in tabular form. Performance statistics, including measures of calibration and discrimination, where available, will be presented alongside their derivation method.

Figures will be used to concisely communicate important information, including (1) summary of the candidate and final predictors included in each model and (2) risk of bias assessment across the four domains (participants, predictors, outcomes and analysis).

Strengths and limitations of the identified models will be considered.

Given our aim is not to compare multiple external validations of a single model, we will not be performing a meta-analysis.[30]

### Patient and public involvement
None.

**Table 3** Information for data extraction and subsequent summary and appraisal

| Domain | Key items |
|---|---|
| Source of data | Source of data (eg, cohort, case–control, randomised trial participants, registry data, etc) |
| Participants | Participant eligibility and recruitment method (eg, location, number of centres, setting, inclusion and exclusion criteria) |
| | Participant description (age, sex, primary VL or relapse case, comorbidities including HIV coinfection) |
| | Details of treatments received |
| | How VL diagnosis is defined (whether consistent for all participants, using serology and/or microscopy, molecular testing, clinical history and physical signs, etc) |
| | Study dates |
| Outcome(s) to be predicted | Type of outcome (eg, single or combined endpoints) |
| | Definition and method for measurement of outcome (for example, is mortality disease-specific or all-cause, is cure/initial failure/relapse diagnosed based on clinical symptoms and/or diagnostic testing) |
| | Was the same outcome definition (and method for measurement) used in all patients? |
| | Time of outcome occurrence or summary of duration of follow-up |
| | Was the outcome assessed without knowledge of the candidate predictors (ie, blinded)? |
| Candidate predictors | Number and type of predictors (eg, demographics, patient history, physical examination, laboratory parameters, HIV status, disease characteristics, etc) |
| | Definition and method for measurement of candidate predictors (including whether defined and measured in a similar way for all participants) |
| | Timing of predictor measurement (eg, at patient presentation, at diagnosis, at treatment initiation or otherwise) |
| | Handling of predictors in the modelling (eg, continuous, linear, non-linear transformations or categorised) |
| Sample size | Number of participants and number of outcomes/events |
| | Events per candidate predictor |
| | Whether the authors describe a sample size calculation |
| Missing data | Number of participants with any missing value (including predictors and outcomes) |
| | Number of participants with missing data for each predictor |
| | Handling of missing data (eg, complete-case analysis, imputation or other methods) |
| Model development | Modelling method (eg, logistic, survival or other) |
| | Modelling assumptions satisfied |
| | Description of participants that were excluded from the analysis with justification |
| | Method for selection of predictors for inclusion in multivariable modelling (eg, all candidate predictors, preselection based on unadjusted association with the outcome) |
| | Method for selection of predictors during multivariable modelling (eg, full model approach, backward or forward selection) and criteria used (eg, p value, Akaike information criterion) |
| | Shrinkage of predictor weights or regression coefficients (eg, no shrinkage, uniform shrinkage, penalised estimation) |
| Model performance | Calibration (calibration plot, calibration slope, Hosmer-Lemeshow test), discrimination (C-statistic, D-statistic, log-rank) and overall performance measures with confidence intervals |
| | Classification measures (eg, sensitivity, specificity, predictive values, net reclassification improvement) and whether a priori cut points were used |
| Model evaluation | Method used for testing model performance: development dataset only (apparent performance, random split of data, resampling methods, eg, bootstrap or cross-validation, none) or separate external validation |
| | For external validations; data source and participants to be described as per 'source of data' and 'participants' domains. Definitions and distributions (including missing data) of outcome and candidate predictors |
| | In case of poor external validation, whether model was updated or extended (eg, intercept recalibrated, predictor effects adjusted, or new predictors added) |
| Results | Final and other multivariable models presented, including predictor weights or regression coefficients, intercept, baseline survival, model performance measures (with SEs or CIs) |
| | Any alternative presentation of the final prediction models, for example, sum score, nomogram, score chart, predictions for specific risk subgroups with performance |

Continued

**Table 3** Continued

| Domain | Key items |
|---|---|
| | Comparison of the definition and distribution of predictors (including missing data) for development and validation datasets |
| Interpretation and discussion | Study authors' interpretation of presented models (intended use, clinical utility, etc) |
| | Study authors' reported strengths and limitations |
| Miscellaneous | Source of funding/sponsor |
| | Any declared conflicts of interest |
| | Methodological guidelines used |

Adapted from CHecklist for critical Appraisal and data extraction for systematic Reviews of prediction Modelling Studies (CHARMS) and Prediction model Risk Of Bias ASsessment Tool (PROBAST).
CI, confidence interval; HIV, human immunodeficiency virus; SE, standard error; VL, visceral leishmaniasis.

## ETHICS AND DISSEMINATION

Ethics approval is not required for this review. The systematic review will be submitted to an open-access journal for peer-review and publication. Findings will also be disseminated through conference presentations, the research group's website (www.iddo.org/research-themes/visceral-leishmaniasis) and social media channels. All extracted information will be made freely available as supplemental material submitted during publication.

## DISCUSSION

We present a protocol for the first systematic review of prognostic models for clinical outcomes in patients diagnosed with VL; a neglected tropical disease that affects some of the most disadvantaged communities in the world. Thoughtful risk stratification of patients using prognostic models can assist clinical decision-making and inform policy, guiding the optimal allocation of often-limited resources. By identifying, summarising and appraising the published VL prognostic models, we hope that the planned systematic review will serve as a comprehensive resource for VL stakeholders, including healthcare workers, policymakers and researchers.

Clinical outcomes in VL are heterogeneous, with rates of initial treatment failure, relapse and mortality varying according to known and unknown factors. Many predictors of poor clinical outcomes have been identified, including extremes of patient age, severity of clinical signs and symptoms, laboratory investigations, the immune status of the patient (including the presence of advanced HIV), the patient's clinical management, geographical location, parasite genotype and resistance profile.[9 31 32] The relative contributions to patient outcomes of these inter-related factors can be described through multivariable modelling and subsequently used to estimate individual patient risk in the form of a prognostic model. However, such a model's performance in a new population can only be directly assessed through external validation. Indeed, this is considered an essential step prior to model use, but infrequently performed in practice.[21] The planned systematic review will concisely summarise key information presented across all identified prognostic model studies. VL healthcare providers and policymakers can then use this information, including performance estimates from external validations, to assess a model's applicability to their own patient population.

This review will not only serve healthcare providers and policymakers in identifying relevant risk stratification tools, but also provide a resource for research groups aiming to validate or update existing prognostic models. The Infectious Diseases Data Observatory is developing a data repository of individual participant data (IPD) from VL clinical trials and observational studies (www.iddo.org/research-themes/visceral-leishmaniasis).[33] A VL data platform including IPD presents an exciting opportunity for the development, validation and updating of prognostic models.[34]

An important strength of the planned review is its broad eligibility criteria; we include models describing all clinical outcomes and impose no restriction on model setting, publication language or prediction horizon (elapsed period between the intended time of model use and the outcome being predicted). Given concerns about study quality and accessibility, we will not be reviewing unpublished nor non-peer-previewed studies, such as conference abstracts or educational theses. A further limitation of the planned review is that we will not be contacting study authors to request unreported information, although we will explicitly report where information is missing. A systematic assessment of the current use and impact of VL prognostic models, including policy guidelines, is considered beyond the scope of the review, however, the review's findings will be considered in the context of current practice as understood by the authors.

In summary, we present a protocol for the systematic review of prognostic models of clinical outcomes for patients diagnosed with VL. With the aim of identifying, summarising and appraising the available risk models, we hope to provide a current reference to stakeholders engaged in VL patient care, policy and research.

**Contributors** The study concept and design were conceived by JW, PJG, KS and PD. EKH developed the search strategy. JW, FC and PD will complete the literature screening. JW, SH and PD will perform data extraction. KS and PD have provided statistical support. FA provided disease-specific expert advice. The first draft of this manuscript was prepared by JW, who will also draft the completed systematic review. All authors critically reviewed and approved the submitted version.

**Funding** JW is undertaking a PhD supported in part by the University of Oxford through the Clarendon Scholarship and Clinician-Investigator Award, and the Infectious Diseases Data Observatory, which is itself supported by a diverse portfolio of grants, including from the Bill & Melinda Gates Foundation (recipient: Professor Philippe Guerin; ref: INV-004713). The funders have no role in the conception, design or conduct of the systematic review.

**Competing interests** None declared.

**Patient and public involvement** Patients and/or the public were not involved in the design, or conduct, or reporting, or dissemination plans of this research.

**Patient consent for publication** Not applicable.

**Provenance and peer review** Not commissioned; externally peer reviewed.

**ORCID iD**
James Wilson http://orcid.org/0000-0003-3615-4928

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
