## [Reviewer comments · BMJ Open]

ARTICLE DETAILS

TITLE (PROVISIONAL)	Prognostic prediction models for clinical outcomes in patients diagnosed with visceral leishmaniasis: protocol for a systematic review
AUTHORS	Wilson, James; Chowdhury, Forhad; Hassan, Shermarke; Harriss, Elinor; Alves, Fabiana; Dahal, Prabin; Stepniewska, Kasia; Guérin, Philippe J.

VERSION 1 – REVIEW

REVIEWER	Lindoso, Jose Angelo Lauletta Universidade de Sao Paulo, Infectious diseases
REVIEW RETURNED	03-Jul-2023

GENERAL COMMENTS	The authors propose a systematic review of prognostic models for visceral leishmaniasis (VL). The purpose of the study is relevant given the scarcity of information on the subject and also the absence of a homogeneous platform that can be used in different regions. As a suggestion, I believe that some points related to the geographic region should be taken into account, since, as is known, the visceral leishmaniasis that occurs in the Indian subcontinent is different from the disease in the Americas. Another point to be considered, when evaluating the results obtained, concerns the population involved in the studies. It is known that the behavior of VL is different between immunosuppressed and non-immunosuppressed individuals. Finally, it is important that, at the end of the systematic review, the authors try to build a VL prognosis model that can be replicated, with its particularities.
--

VERSION 1 – AUTHOR RESPONSE

Reviewer: 1

Dr. Jose Angelo Lauletta Lindoso, Universidade de Sao Paulo, Instituto de Infectologia Emilio Ribas
Comments to the Author:

The authors propose a systematic review of prognostic models for visceral leishmaniasis (VL). The purpose of the study is relevant given the scarcity of information on the subject and also the absence of a homogeneous platform that can be used in different regions.

As a suggestion, I believe that some points related to the geographic region should be taken into account, since, as is known, the visceral leishmaniasis that occurs in the Indian subcontinent is different from the disease in the Americas.

Thank you for highlighting this important point. We have added a paragraph to the Discussion section (second paragraph) that considers the different factors known to predict clinical outcomes, including geography and patient immunity. This paragraph also provides some important context that highlights to the reader how prognostic models can communicate important information about their use.

Another point to be considered, when evaluating the results obtained, concerns the population involved in the studies. It is known that the behavior of VL is different between immunosuppressed and non-immunosuppressed individuals.

Thank you – we had originally touched on this in the Introduction section. In the Discussion section (described above) we have explicitly referenced HIV as a course of poor outcomes.

Finally, it is important that, at the end of the systematic review, the authors try to build a VL prognosis model that can be replicated, with its particularities.

We certainly agree with this. Hopefully the systematic review will provide a good starting point by telling us what is already available. Perhaps the individual patient data at IDDO data platform can be used to update a model that already exists.

*** **

COI statement(s):

Reviewer: 1

Competing interests of Reviewer: None.